# OpenReview forum: "Finite-time Convergence Analysis of Actor-Critic with Evolving Reward"
_ICML.cc/2026/Conference — ICML 2026 regular_

### Official Review · Reviewer_qTnm · 2026-03-02

**Soundness:** 3
**Presentation:** 4
**Significance:** 3
**Originality:** 3
**Overall Recommendation:** 4
**Confidence:** 4

**Summary:**

This paper provides an analysis of single-timescale actor-critic algorithm with an evolving reward function under Markovian sampling. They show with slow-evolving conditions on the reward function, the convergence rate can match the one for the traditional static reward function.

**Compliance With Llm Reviewing Policy:**

Affirmed.

**Final Justification:**

Addressed my concern; therefore, I keep my positive score.

**Key Questions For Authors:**

See W2, if the authors can prove its correctness, I am willing to increase my score.

**Limitations:**

Yes

**Strengths And Weaknesses:**

Strength: This paper is well-written and easy to follow, the problem it aims to solve is clear and the motivation is meaningful, and the related work section is comprehensive. Moreover, Proposition 4.13 provides a novel analysis of the effects of Markovian noise. It does not require a $\tau$-step traceback; therefore, it improves the convergence bound by $O(\tau^2)$.

I have several questions, not necessarily to be weaknesses.

Weakness:

1. This paper adopts Chen 2025[2]'s setting that the sampling scheme requires a simulator that can reset to the initial distribution $\rho$. Particularly, at the end of Section 3.2, the authors say "$\hat{P}$ ensures ergodicity". My question is, since your analysis does not rely on the mixing time of the ergodic Markov chain, why do you adopt this setting instead of using the original $P(\cdot|s,a)$(i.e., the setting adopted by [1])? My educational guess is if it works for the mixed transition probability dynamics, it should work for the original pure transition probability dynamics. Can authors comment on this?

2. Although this paper shows an improvement of order $O(\log^2 T)$, I suspect the correctness. For example, for $I_3$ in equation (17), $\bar{h}\_{\theta}(\theta\_t,\omega\_t,\phi\_t, \hat{\nu}\_t)$ is a function of the parameters $\theta\_t,\omega\_t,\phi\_t, \hat{\nu}\_t$, and these parameters are actually functions of all the random variables in the past history, that is $s\_0, a\_0, s\_1,...$, therefore, I suspect line 911 is not automatically true because it should be an integration over the joint probability distribution of all past random variables instead of just the tuple $(s\_t,a\_t,s\_t')$. Similar problems exist for other Markovian noises.

[1] Chen, X. and Zhao, L. Finite-time analysis of  single
 timescale actor-critic. Advances in Neural Information
Processing Systems, 2023.

[2] Chen, X. and Zhao, L. On the convergence of continuous single-timescale actor-critic. In Forty-second International Conference on Machine Learning, 2025.

---

> ### Author Rebuttal · Authors · 2026-03-31
>
> Thank you for your time and effort in reviewing our paper! We are grateful for your constructive suggestions, which have significantly guided our improvements. Please find our responses to your comments below.
>
> **Q1**: This is a great question! Our analysis on Markovian noise does not apply to the original transition kernel $P$ because our analysis relies on the contractivity of $\hat{P}$, which is not generally possessed by $P$. Specifically, under the assumption of uniform ergodicity, there exist constants $c>0$ and $\kappa\in(0,1)$ such that $\|| \mathcal{P}^\tau \mu- \mu^* \||_1 \leq c\kappa^\tau$ for any $\tau\geq0$. In contrast, our analysis utilizes the property that $\|| \hat {\mathcal{P}} \nu-\nu^* \||_1\leq\gamma$. Contractivity implies ergodicity with $c=1$ and $\kappa=\gamma$, but in general, ergodicity does not imply contractivity (for instance, one can have $c>1$). Therefore, adopting the mixed transition kernel $\hat {\mathcal{P}} $ together with our novel analysis contributes to the $O(\log^2T)$ improvement.
>
> **Q2**: We apologize for the misleading statements in our proof, but we respectfully clarify that our conclusion remains valid.
>
> In Eq. 17 (Page 16), we decompose the first-order terms of $\mathbb{E}[J_{\varphi_t}(\theta_{t+1})]-J_{\varphi_t}(\theta_t)$ into $I_1$, $I_2$, and $I_3$. The expectation is conditioned on the history up to time $t$, i.e., over the sampled transition $(s_t,a_t,s_t')$. This expectation is then absorbed into $\bar h_{\theta}(\cdot)$, which appears in $I_1$, $I_2$ and $I_3$. Therefore, line 913 is valid for $I_3$, and line 915 holds for $\mathbb{E}[I_3]$ according to Prop. 4.13, where the expectation is taken over the history. We can still apply this inequality after summing over iterations and taking the expectation over history to derive Equation 18 (Page 17). The same reasoning applies to $J_3$, and the theorem is not affected. We will revise the presentation in the final version.
>
> We hope our response addresses your concerns. If so, we wonder if you could kindly consider raising your score? We will also be happy to answer any further questions you may have. Thank you very much!

---

> > ### Author Rebuttal · Reviewer_qTnm · 2026-03-31
> >
> > The authors explained the necessity of adopting the mixed transition kernel $\hat {\mathcal{P}} $, which, instead of relying on the traditional mixing time analysis, exploits the contraction property of the mixed transition kernel. Therefore, it makes sense to me that they have improved the final convergence bound, and I have decided to increase the score.

---

### Official Review · Reviewer_fEQv · 2026-03-06

**Soundness:** 3
**Presentation:** 3
**Significance:** 3
**Originality:** 2
**Overall Recommendation:** 5
**Confidence:** 4

**Summary:**

The paper studies the sample complexity of actor-critic algorithm to a stationary point, under evolving reward. The question is important with various application such as life-long learning, etc.  The establishes a concrete theoretical result.

**Compliance With Llm Reviewing Policy:**

Affirmed.

**Key Questions For Authors:**

Q1) Assumption 4.1 is made in almost all the work. To my best of knowledge, this assumption requires Bellman operator to be contraction is L2 norm, instead of L_infy norm. Can authors comment, when this Assumption 4.1 is satisfied ?


Q2) The paper considers evolution of reward which is also important. What do authors think about evolving transition kernel case. How can existing results can translate to that case, what are the expected challenges.

Q3) The analysis is similar to [Chen, X. and Zhao, L. 2023], can author outline the major  differences and challenges.

Q4) The considers the scenario where reward is continuously evolving without any restriction. In other words,  consider the setting, where reward can evolve but always limited to set as " A car never becomes a aeroplane". Adding this restriction, how can the result be improved? Maybe using reward robustness [5] trick help? What do authors think about this?

[5] Gadot, U., Derman, E., Kumar, N., Elfatihi, M. M., Levy, K., & Mannor, S. (2024). Solving Non-rectangular Reward-Robust MDPs via Frequency Regularization. Proceedings of the AAAI Conference on Artificial Intelligence, 38(19), 21090-21098. https://doi.org/10.1609/aaai.v38i19.30101

**Limitations:**

Yes.

**Strengths And Weaknesses:**

Strengths:
S1) Concrete theoretical result.


Weakness:
W1)  The result establish local convergence with complexity of $O(\epsilon^{-2})$ which translates to global complexity of $O(\epsilon^{-4})$ [1]. The work [1], has establish $O(\epsilon^{-3})$ global complexity in the static reward case.  How does the existing result translates to the global convergence sense?




Minor:
1)  Sub-section : "Finite-time analysis of Actor-Critic Methods": Missing mention of state-of-the-art global convergence of actor critic [1].
2) It should be mentioned:  Line 185:  Under function approximation even in linear cases can lead to undesired behviours such oscillations or sub-optimal convergence [2].






[1] @inproceedings{
kumar2025on,
title={On the Convergence of Single-Timescale Actor-Critic},
author={Navdeep Kumar and Priyank Agrawal and Giorgia Ramponi and Kfir Yehuda Levy and Shie Mannor},
booktitle={The Thirty-ninth Annual Conference on Neural Information Processing Systems},
year={2025},
url={https://openreview.net/forum?id=OixkI1jSZD}
}

[2]@misc{eshwar2025monotoneconservativepolicyiteration,
      title={Monotone and Conservative Policy Iteration Beyond the Tabular Case},
      author={S. R. Eshwar and Gugan Thoppe and Ananyabrata Barua and Aditya Gopalan and Gal Dalal},
      year={2025},
      eprint={2506.07134},
      archivePrefix={arXiv},
      primaryClass={cs.LG},
      url={https://arxiv.org/abs/2506.07134},
}

---

> ### Author Rebuttal · Authors · 2026-03-31
>
> Thank you for your time and effort in reviewing our paper! We are grateful for your positive and constructive suggestions, which have significantly guided our improvements. Please find our responses to your comments below.
>
> **W1**:
>
> According to the gradient domination lemma (Mei et al., 2022), our local convergence rate of $O(\epsilon^{-2})$ translates to a global convergence rate of $O(\epsilon^{-4})$. However, this translation requires additional assumptions regarding the parameterization of the policy. Kumar et al. (2025) demonstrated that it is possible to achieve a global convergence rate of $O(\epsilon^{-3})$ in the tabular case by exploiting the gradient domination structure. However, it remains unresolved whether this result applies to the continuous state-action space considered in this paper.
>
> [1] Mei, J., Xiao, C., Szepesvári, C., and Schuurmans, D. On the Global Convergence Rates of Softmax Policy Gradient Methods. ICML 2020.
>
> [2] Kumar, N., Agrawal, P., Ramponi, G., Levy, K.Y., & Mannor, S. On the Convergence of Single-Timescale Actor-Critic. NeurIPS 2025.
>
> **Q1**:
>
> We are unsure if we fully understand your question, but we can provide an intuitive explanation. Consider a finite state space with a one-hot feature $\phi$. In this context, $\lambda_\Sigma$ represents the minimum visitation probability across all states and all possible policies. Consequently, Assumption 4.1 ensures that the policy can cover the entire state space, which can be achieved by guaranteeing a non-zero probability of selecting any given action. For an infinite state space, Assumption 4.1 can then be translated to the coverage of the feature space.
>
> **Q2**:
>
> Thank you for highlighting this. We believe that a similar analysis can be transferred to the evolving transition kernel case. The key steps would be establishing the Lipschitz continuity of the actor objective $J$ and the optimal critic parameter $\omega^*$ with respect to the evolving transition kernel $P$.
>
> **Q3**:
>
> Our analysis is inspired by Chen & Zhao's[3]. The main differences and challenges are two-fold.
> First, we introduce a reward parameter $\varphi$ to handle the evolving reward, while they use a reward estimator to handle the infinite-horizon average-reward setting. This difference arises from the distinct primal settings, which require entirely different technical tools. The estimated reward does not alter the optimal policy, while the evolving reward does have an impact. This necessitates additional control over the changes in the objective function $J$ and the optimal critic $\omega^*$ against the variations in the reward parameter $\varphi$ during the iterations.
> Second, we employ a novel analysis of the Markovian noise that results from approximating the discounted state visitation $\nu$ with a mixed kernel $\hat{P}$. Our analysis relies on the $\gamma$-contractivity of $\hat{P}$, whereas their analysis relies on the ergodicity of the original kernel $P$. This distinction leads to an improvement in the final convergence bound by a factor of $O(\log^2T)$.
>
> [3] Chen, X., & Zhao, L. Finite-time analysis of single-timescale actor-critic. NeurIPS 2023.
>
> **Q4**:
>
> This is a great question! However, directly incorporating the additional restrictions on the reward directly into our analysis is quite challenging. For instance, consider the case that the reward function oscillates between two distinct points, the reward variation $F_T$ will be $\Omega(1)$, which implies divergence of the algorithm according to our main theorem. However, if we have some prior knowledge of the reward function, it will be an interesting future direction to consider the evolving belief of the reward.
>
> **Minor**:
>
> Thank you for pointing out the missing references. We will include them in the revision.
>
> We hope our response addresses your concerns. If so, we wonder if you could kindly consider raising your score? We will also be happy to answer any further questions you may have. Thank you very much!

---

> > ### Author Rebuttal · Reviewer_fEQv · 2026-04-02
> >
> > Thanks for the response. I will keep my original score.

---

### Official Review · Reviewer_DimS · 2026-03-09

**Soundness:** 1
**Presentation:** 3
**Significance:** 1
**Originality:** 1
**Overall Recommendation:** 2
**Confidence:** 3

**Summary:**

The paper studies an actor–critic algorithm with an evolving reward function and provides a convergence analysis under Markovian sampling that recovers existing rates from prior work. In addition, the authors analyze the distribution mismatch term and improve the best known rate by removing a $\log^2 T$ factor.

**Compliance With Llm Reviewing Policy:**

Affirmed.

**Final Justification:**

I respectfully disagree that it is sufficient to state that “the analysis is based on standard assumptions and is rigorously derived.” Without empirical evaluation, it is difficult to assess how strong or realistic these assumptions are, when they may fail in practice, and what the overall contribution of the work is. Therefore, I maintain my score.

**Key Questions For Authors:**

- The authors claim: “Many popular practical reinforcement learning (RL) algorithms employ evolving reward functions, through techniques such as reward shaping, entropy regularization, or curriculum learning, yet their theoretical foundations remain underdeveloped.” Is this claim accurate? For example, policy gradient (PG) methods with KL regularization have already been studied in the linear function approximation setting. How does this work relate to those existing results?
- “Introducing an entropy or KL regularization term to the optimization objective, which is equivalent to modifying the reward according to the current policy.” If this statement is true, and given that KL-regularized PG methods have already been studied in the linear function approximation setting, what is the contribution of this work?
- “In addition, we introduce a novel analysis on the distribution mismatch caused by Markovian sampling, improving the convergence rate by a factor of $\log^2 T$ in the static-reward case.” What is the impact of this contribution? Is removing a $\log^2 T$ factor considered significant, even from a theoretical perspective?
- “While we focus on entropy regularization for clarity, our analysis also applies to KL regularization against a fixed reference policy $\pi_{ref}$.” However, in practice the KL regularization is not taken with respect to a fixed $\pi_{ref}$, but rather against $\pi_t$, which changes after every interaction with the environment. Does $\pi_{ref}$ change in your analysis? Am I misunderstanding the claim?

**Limitations:**

yes

**Strengths And Weaknesses:**

Strengths:
- The paper is well written. The authors clearly describe the motivation and the technical machinery required to understand their contributions before presenting the main theoretical results. This helps make the development of the theory easier to follow.

Weaknesses:
- The contribution appears limited. While it is valuable to formalize the practical setting of evolving reward functions, it is not clear what key insight the paper provides or how the broader community would benefit from the results. For example, it is unclear how these findings would influence practitioners or theorists in their future work. I suggest that the authors further develop and better communicate the implications of their theoretical findings and validate these insights through experiments.

---

> ### Author Rebuttal · Authors · 2026-03-31
>
> Thank you for your time and effort in reviewing our paper! We are grateful for your constructive suggestions, which have significantly guided our improvements. Before addressing your specific questions, we would like to clarify the novelty and generality of our evolving-reward framework and the contributions of this paper.
>
> As stated in Section 3.3, a regularized reward can be parameterized as $$\tilde r_{\varphi,\theta}(s, a)=r(s, a;\varphi)-\alpha(\varphi)\log\pi_\theta(a|s).$$ Its evolution arises from three components: (i) the base reward $r(s, a)$, (ii) the regularization coefficient $\alpha$, and (iii) the current policy $\pi_\theta$. Within this framework, practical techniques such as reward shaping methods (e.g., random network distillation (RND)) and curriculum learning primarily affect component (i). Adaptive entropy regularization relates to components (ii) and (iii), while adaptive KL regularization involves all three components (i), (ii), and (iii) if both $\pi_{ref}$ and $\alpha$ change during training.
>
> To the best of our knowledge, existing literature on PG or AC on regularized MDP all assume a static base reward and a fixed $\alpha$. This means they only consider component (iii) while neglecting the dynamic nature of the first two components that impact the reward. However, these first two components are crucial because their evolution can alter the optimal policy. This oversight raises concerns about the convergence of RL algorithms when applying the aforementioned techniques.
>
> To address this essential issue, we focus on the first two components and introduce a unified parameter, $\varphi$, to capture their evolution. We demonstrate that the long-term errors of the actor and the critic can both be bounded in terms of reward variation. Furthermore, we provide sufficient conditions for the "evolving rate" of the reward to ensure convergence.
>
> For practitioners, this work not only lays a solid theoretical foundation for a wide range of commonly used techniques but also offers a systematic principle for designing evolving rewards. For theorists, this paper introduces a novel framework for understanding evolving rewards, a unique technical tool for analyzing changing objectives, and an improved analysis of Markovian noise, all of which are valuable for future research.
>
> Below, we briefly answer your questions.
>
> **Q1.1**: We will revise the presentation to emphasize **adaptive** regularization in our revision, as "adaptive" refers to the evolving $\alpha(\varphi)$, which has not been theoretically examined by existing literature but falls within the scope of our study.
>
> **Q1.2**: This study is the first to analyze the finite-time convergence of an actor-critic algorithm within an evolving reward framework. When it is reduced to the static-reward scenario, either with or without entropy regularization, our findings yield comparable results.
>
> **Q2**: Our contribution consists of three main parts. Firstly, we formalize the important problem of Actor-Critic with Evolving Reward. Second, we present a novel non-asymptotic convergence result for the algorithm under standard assumptions, which supports a variety of practical RL techniques, including adaptive reward shaping, adaptive entropy/KL regularization, and curriculum learning. Finally, we develop useful technical tools for analyzing both the evolution of the system and the effects of Markovian noise.
>
> **Q3**: Although the improvement of the $\log^2T$ factor can be considered as a secondary contribution of this paper, the key propositions and analyses we developed in this work to achieve this improvement in Markovian sampling are applicable to other works that use Markovian sampling as well (e.g. [1][2][3]). This could lead to similar enhancements and may benefit future studies in this area.
>
> **Q4**: As mentioned in Section 3.1, the term regarding $\pi_{ref}$ is absorbed into the base reward $r(s, a;\varphi)$ and is subsequently reparameterized by $\varphi$. When $\pi_{ref}=\pi_t$, this can be rewritten as $\varphi_t=\theta_t$. Note that $\varphi_t$ adopts a gradient-based update and the learning rate is of the same order as the actor's, then according to Corollary 4.12, an $O(1/\sqrt{T})$ convergence can be obtained in this case.
>
> We hope our response addresses your concerns. If so, we wonder if you could kindly consider raising your score? We will also be happy to answer any further questions you may have. Thank you very much!
>
> [1] Tengyu Xu, Zhe Wang, Yingbin Liang: Non-asymptotic Convergence Analysis of Two Time-scale (Natural) Actor-Critic Algorithms. CoRR abs/2005.03557 (2020)
>
> [2] Haoxing Tian, Alex Olshevsky, Yannis Paschalidis: Convergence of Actor-Critic with Multi-Layer Neural Networks. NeurIPS 2023
>
> [3] Xuyang Chen, Lin Zhao: Finite-Time Analysis of Single-Timescale Actor-Critic. NeurIPS 2023

---

> > ### Author Rebuttal · Reviewer_DimS · 2026-04-03
> >
> > Thank you for the response. My primary concern remains the lack of empirical evidence. I find empirical evaluation necessary to assess the validity of the claims and assumptions (both in simple settings that closely match the theory and in more practical environments) to understand the significance of the contribution. Without such evaluation, I find the contribution too narrow and therefore maintain my score.

---

> > > ### Author Response · Authors · 2026-04-05
> > >
> > > Thank you for your feedback.
> > >
> > > We would like to note that, to our knowledge, empirical evaluation is not considered a necessary component for papers in this area. In fact, a large portion of our related works on the convergence analysis of PG and AC (to list some of them, [1-9]) do not include empirical evaluation in the paper.
> > >
> > > In relation to our paper, we respectfully disagree with the necessity of empirical evaluation. The motivation of this paper is to close the theoretical gap in the analysis of practical RL algorithms by incorporating the concept of evolving reward into the theoretical framework. This concept arises from many common practices but has never been formalized for rigorous analysis. The main contribution of this paper to practical RL is validating existing methods, providing a theoretical understanding of how these methods work and why they work well. In this context, we believe that while additional empirical evaluation could possibly enhance the paper, it is not essential for the following reasons: (i) the analysis is based on standard assumptions and is rigorously derived, and (ii) evolving-reward techniques have been widely used and have proven to be effective in practice.
> > >
> > > We would really appreciate it if you could reconsider raising the score of our paper. Please let us know if you have any further questions. We will be happy to address them. Thank you very much.
> > >
> > > [1] Alekh Agarwal, Sham M. Kakade, Jason D. Lee, and Gaurav Mahajan: On the theory of policy gradient methods: optimality, approximation, and distribution shift. JMLR, 2021.
> > >
> > > [2] Jincheng Mei, Chenjun Xiao, Csaba Szepesvári, and Dale Schuurmans: On the global convergence rates of softmax policy gradient methods. ICML, 2020.
> > >
> > > [3] Lin Xiao: On the convergence rates of policy gradient methods. JMLR, 2022.
> > >
> > > [4] Yuhao Ding, Junzi Zhang, Javad Lavaei: On the global optimum convergence of momentum-based policy gradient. AISTATS, 2022.
> > >
> > > [5] Mondal, Washim Uddin, Vaneet Aggarwal: Improved Sample Complexity Analysis of Natural Policy Gradient Algorithm with General Parameterization for Infinite Horizon Discounted Reward Markov Decision Processes. PMLR, 2024.
> > >
> > > [6] Alex Olshevsky, Bahman Gharesifard: A Small Gain Analysis of Single Timescale Actor Critic. SIAM J. Control. Optim. 2023.
> > >
> > > [7] Haoxing Tian, Alex Olshevsky, Yannis Paschalidis: Convergence of Actor-Critic with Multi-Layer Neural Networks. NeurIPS, 2023.
> > >
> > > [8] Xuyang Chen, Lin Zhao: Finite-Time Analysis of Single-Timescale Actor-Critic. NeurIPS, 2023.
> > >
> > > [9] Xuyang Chen, Lin Zhao: On the convergence of continuous single-timescale actor-critic. ICML, 2025.

---

### Decision · Program_Chairs · 2026-04-30

**Decision:**

Accept (regular)

**Comment:**

The paper considers reinforcement learning (RL) settings with changing reward functions, where the changes are deliberate, rather than being imposed, as in the context of reward shaping, entropy regularization, and curriculum learning. In these settings, it analyzes the convergence of a single-timescale actor-critic algorithm to a stationary point and establishes a finite-time guarantee that matches that of the static reward, in terms of dependence on the number of iterations, under a slow rate of reward change.

The paper presents a novel theoretical contribution in the well-motivated setting of RL with deliberately changing reward function. It sets several assumptions for establishing the convergence guarantee, most of which are commonly used in this type of contribution; however, it would still be important and helpful to provide intuition, justification, and practical consideration for them to the extent possible. Related to that, it would be better to move the remark on Assumption 4.8 under Proposition 4.10 to right after the statement of that assumption. The titles selected for some assumptions and propositions do not properly capture their content. The technical steps and analysis have considerable overlap with the two cited papers by Chen & Zhao, 2023 & 2025, therefore, limiting the novelty and significance. While differences exist, particularly the changing reward setting and the improved Markovian sampling rate mentioned in the authors' rebuttal as well, it is imperative to clearly state the similarities and differences in the paper.